# Perceptions of Quality of Interprofessional Collaboration, Staff Well-Being and Nonbeneficial Treatment: A Comparison between Nurses and Physicians in Intensive and Palliative Care

**DOI:** 10.3390/healthcare12060602

**Published:** 2024-03-07

**Authors:** Daniel Schwarzkopf, Frank Bloos, Winfried Meißner, Hendrik Rüddel, Daniel O. Thomas-Rüddel, Ulrich Wedding

**Affiliations:** 1Center for Sepsis Control and Care, Jena University Hospital, 07747 Jena, Germany; frank.bloos@med.uni-jena.de (F.B.); hendrik.rueddel@med.uni-jena.de (H.R.); daniel.thomas@med.uni-jena.de (D.O.T.-R.); 2Department of Anesthesiology and Intensive Care Medicine, Jena University Hospital, 07747 Jena, Germany; winfried.meissner@med.uni-jena.de; 3Department of Palliative Medicine, Jena University Hospital, 07747 Jena, Germany; ulrich.wedding@med.uni-jena.de; 4Comprehensive Cancer Center Central Germany (CCCG), 07743 Jena, Germany

**Keywords:** patient care team, critical care, palliative care, job satisfaction, occupational stress, cross-sectional survey

## Abstract

This study assessed differences in interprofessional collaboration, perception of nonbeneficial care, and staff well-being between critical care and palliative care teams. In six German hospitals, a staff survey was conducted between December 2013 and March 2015 among nurses and physicians in intensive and palliative care units. To allow comparability between unit types, a matching was performed for demographic characteristics of staff. N = 313 critical care and 79 palliative care staff participated, of which 72 each were successfully matched. Critical care nurses perceived the poorest overall quality of collaboration compared with critical care physicians and palliative care physicians and nurses. They also reported less inclusive leadership from attendings and head nurses, and the least collaboration on care decisions with physicians. They were most likely to perceive nonbeneficial care, and they reported the lowest levels of job satisfaction and the highest intention to leave the job. In partial correlations, aspects of high-quality collaboration were associated with less perceived nonbeneficial care and higher staff well-being for both critical care and palliative care staff. Our findings indicate that critical care teams could improve collaboration and enhance well-being, particularly among nurses, by adopting principles of collaborative work culture as established in palliative care.

## 1. Introduction

Interprofessional/interdisciplinary collaboration in health care has been described as a process in which professionals from different disciplines and professions work together on the basis of shared decision-making, mutual trust and respect, and open and effective communication to jointly provide qualified patient care [1,2]. In both critical care and palliative care, patients with life-threatening illnesses are treated by interprofessional care teams. Although the primary goals and environments of critical care and palliative care are different, the teams face similar social demands as they are regularly confronted with the dying and suffering of patients, and the emotional needs of patients’ relatives in highly stressful situations.

High-quality collaboration is associated with improved patient outcomes and healthcare worker well-being. For example, intensive care unit (ICU) nurses’ perceptions of good nurse-physician collaboration about transfer decisions were associated with a lower risk of patient mortality or readmission to ICU [3]; nurses’ ratings of nurse-physician relations as well as nurse management were associated with less experience of burnout, higher job satisfaction, and improved ratings of quality of care [4]; improved collaboration in the interprofessional team in a surgical unit was associated with improved quality of care and postoperative functioning of patients, as well as reduced pain and length of stay [5]; improved ratings of nurse-physician collaboration by nurses on surgical and medical wards were associated with reduced length of stay for patients [6]; and a review of the literature on team approaches to palliative care concluded that communication, leadership skills and mutual respect are key to the success of multidisciplinary teams [7].

While intensivists have called for improved collaboration to achieve quality [8,9,10], studies have found low levels of nurse-physician collaboration with nurses rating the quality of collaboration significantly lower than physicians [11,12]. Palliative care has a long tradition of emphasizing the interprofessional/interdisciplinary team and is considered a working model for successful collaboration [13,14]. Therefore, it can be expected that palliative care teams would have generally higher ratings of collaboration with less variation across professions. Accordingly, two survey studies of hospice teams in the United States found no differences between the professions (e.g., nurses, physicians, social workers, and chaplains) in their ratings of the quality of interdisciplinary collaboration [13,15]. Except for small single-center studies, little is known about interprofessional/interdisciplinary collaboration in inpatient palliative care units (PCUs) [16,17]. We conducted a single-center study comparing experiences in the context of end-of-life decision-making experiences between ICU and PCU staff and found that PCU staff gave better ratings of the interaction within the team, and while ICU nurses rated interaction poorer than ICU physicians, ratings of PCU nurses and PCU physicians did not differ [16].

Both palliative and intensive care professionals face challenging ethical decisions and have to deal with death and dying, making them vulnerable to moral distress and burnout [18,19,20,21,22]. High-quality collaboration is believed to improve end-of-life decision-making and care in the ICU and prevent the provision of nonbeneficial care to patients [10,14,23]. The provision of nonbeneficial care at the end of life causes unnecessary risk and suffering for patients and families, as well as a waste of healthcare resources [24]. Perceiving nonbeneficial treatment is a major source of moral distress and is associated with burnout and intention to leave the job among critical care workers [25,26,27]. A nationwide Portuguese study found higher rates of burnout among ICU staff compared to staff of PCU, but did not assess differences in the quality of collaboration [22]. Nothing is known about the perception of nonbeneficial treatment among palliative care workers.

Based on the cited literature, we hypothesize that the quality of collaboration is an important predictor of staff well-being and prevention of nonbeneficial treatment in both critical care and palliative care. We further hypothesize that palliative care teams show better quality of collaboration than critical care teams, with less interprofessional variation in ratings. To investigate this, we conducted a multicenter survey study among ICU staff and PCU staff between December 2013 and March 2015.

## 2. Materials and Methods

### 2.1. Study Design

This is a prospective observational study conducted by a paper-pencil survey of nurses and physicians in both ICUs and PCUs in a convenience sample of six German hospitals. To adjust for differences in demographic characteristics between ICU and PCU staff, a 1:1 matching was performed within each participating hospital.

### 2.2. Sample and Procedure

This is a substudy of a cluster randomized controlled trial, which aimed to improve acute sepsis care (Medical Education for Sepsis Source Control and Antibiotics, MEDUSA, ClinicalTrials.gov identifier NCT01187134) [28]. All participating hospitals were invited to take part in a survey of ICU staff, the results of which have been reported previously [27]. Twenty-five of the MEDUSA trial hospitals had both a PCU and an ICU and were invited to participate in a paper-pencil survey of both palliative care and critical care nurses and physicians; six of these hospitals participated in the survey. Experienced senior physicians or department heads served as local study coordinators and distributed the paper-pencil survey to the unit staff between December 2013 and March 2015. All physicians and nurses working in the respective PCUs and ICUs were eligible to participate. Each local study coordinator received feedback on their unit’s participation rate after two and four weeks, and questionnaires were distributed to staff a second time after four weeks.

### 2.3. Measurements

#### 2.3.1. Development of the Questionnaire

The questionnaire was developed in a multi-step process. First, a theoretical framework was developed that included three domains: (a) perceptions of the quality of collaboration in the unit, (b) relevant outcomes affected by the quality of collaboration (staff well-being, perception of nonbeneficial care), and (c) relevant covariates (demographic characteristics, workload). Second, since there was no single validated questionnaire assessing all relevant constructs, we conducted a literature search to select relevant, validated scales or items from existing instruments. Item wordings were partly adapted to the setting of the study (e.g., reference to nurses/physicians or the unit instead of co-workers or the company); English items were translated into German by forward and backward translation. There was no validated scale to measure the perception of nonbeneficial treatment. Therefore, we developed a new scale. This initial pool of items was pre-tested by cognitive interviewing with nurses and physicians from critical and palliative care. The reliability and factorial validity of the scales were tested in the larger sample of 23 ICUs and were reported previously [27]. To ensure reliability of the scales also among PCU staff, we reassessed it in the sample of PCU nurses and physicians.

#### 2.3.2. Content of the Questionnaire

(a) *Perception of quality of collaboration* was assessed using three scales. First, to assess collaboration with a wide range of different professional roles (nurses, head nurses, residents, attendings, consulting physicians, occupational or physiotherapists, and psychologists or social workers), individual items taken from the Safety Attitudes Questionnaire were used (“Describe the quality of collaboration and communication you have experienced with…” using a rating scale from 1-“very low” to 5-“very high”) [29]. Resembling an existing approach, the average of the ratings was used to receive an overall quality of collaboration index [5]. Second, an adapted version of the *Collaboration about Care Decisions scale* (shortened from six to four items) was used to assess nurse-physician collaboration in more detail [30,31]. Because leader behaviors are essential in shaping the internal dynamics of a team, we assessed leadership by head nurses and attending physicians as a third component. We specifically chose to assess *inclusive leadership* because it involves inviting and valuing the contributions of others, leading to an atmosphere of mutual respect between different professions [32]. It was measured using a published scale previously used among critical care staff [32]. We adapted the wording to refer to the leadership either by attending physicians or head nurses. Leadership by head nurses was only assessed by nurses.

(b) *Outcomes possibly affected by quality of collaboration:* In the absence of a validated scale to measure the *perception of nonbeneficial treatment*, we developed a 5-item scale (e.g., “For the patients you treat on your unit: how often do you perceive that …continued life-sustaining treatment unnecessarily prolongs a patient’ suffering?”). We partly adapted existing questionnaire items [25,33]. Scale development has been described previously [27]. Staff well-being was addressed by assessing burnout, job satisfaction, and intention to leave the job. *Burnout* was operationalized by its central quality, emotional exhaustion, measured by the corresponding subscale of the Maslach Burnout Inventory—General Survey [34] (German translation based on an earlier translation [35], received by personal communication with J. Glaser, 18 July 2011 [36]). *Job satisfaction* was measured by a single item, previously used in the context of critical care, indicating satisfaction by one of seven pictures of a face (1 = very angry face, 7 = happy smiling face) [37]. *Intention to leave the job* was measured by a 3-item short form of the Turnover Intentions Scale [38,39].

(c) *Covariates*: Excessive *workload* is the most important predictor of impaired staff well-being [40]. Therefore, we assessed it both to describe the respective work environments in critical care and palliative care and to use it as a covariate in investigating relationships between aspects of collaboration and outcomes affected by it. A three-item scale previously used in the critical care setting was used to measure it [25]. In addition, it is well documented that demographic characteristics like age, gender, job experience, and professional role influence perceptions of teamwork, job stress and satisfaction, and perception of quality of patient care or nonbeneficial care [11,12,19,27]. Therefore, we assessed demographics and used them to match PCU and ICU staff, and as covariates.

Items on workload, collaboration about care decisions, inclusive leadership, and intention to leave the job used a 7-point Likert scale (1-“strongly disagree” to 7-“strongly agree”), and items on perception of nonbeneficial treatment and emotional exhaustion used a 6-point frequency scale (1-“never” to 6-“very often”). All items are shown in the Appendix A.

### 2.4. Statistical Analysis

Cronbach’s alpha was used to estimate the reliability of the scales in both the PCU and ICU samples. To adjust comparisons between PCUs and ICUs for demographic differences, matching was performed within each hospital for the variables of professional role (head nurse, nurse, attending, resident), gender, age (<40 or ≥40 years), and medical experience (<5 or ≥5 years) using a genetic matching algorithm and indicator coding of missing values [41]. The balance of variables between the matched samples was checked by standardized differences [42].

Differences between the four groups ICU nurses, ICU physicians, PCU nurses, and PCU physicians regarding the measured items and scales were graphically presented using boxplots. Overall differences between the four groups were tested using Kruskal–Wallis tests; comparisons between individual groups were executed by the Wilcoxon rank-sum test adjusted for multiple comparisons using the Holm method.

Associations of measures of interdisciplinary collaboration with the perception of nonbeneficial treatment and measures of staff well-being were analyzed within the subgroups of palliative and intensive care staff using partial correlations controlling for occupation (nurse vs. physician), gender, age (<40 years, ≥40 years), and workload. Missing data were handled by pairwise deletion, all tests were conducted at a significance level of α ≤ 0.05, and analyses were performed using the statistical software R, version 4.2.2 [43].

## 3. Results

### 3.1. Participants

Figure 1 shows the study flow chart. Of the 25 invited hospitals, 6 participated with both a PCU and an ICU (Figure 1). Of these, three were university hospitals and three were primary care hospitals. Appendix A shows the characteristics of the participating hospitals and units. The number of beds ranged from 7 to 12 for PCUs and from 7 to 58 for ICUs. Occupational/physical therapists and social workers were integrated into all six PCU teams, and psychologists were integrated into four teams. Occupational/physical therapists were integrated into three of the six participating ICU teams, social workers were integrated into none, and psychologists were integrated into two teams.

Table 1 shows the demographic characteristics of the participating staff. Participating palliative care staff were older, had more years of general medical experience and fewer years of specialized professional experience (in intensive care or palliative care, respectively), and were more often in a leadership position compared to intensive care staff (all *p* ≤ 0.01). After matching for gender, age, professional role, and medical experience, these characteristics did not differ significantly between groups (all *p* ≥ 0.262). Matching reduced the standardized differences from a maximum of 1.08 to a maximum of 0.12 (Appendix A). Matching for specialized experience in critical care vs. palliative care, respectively, was not possible because the differences were too large, and therefore, significant differences remained after matching (*p* ≤ 0.001). Differences in the perceived workload are presented in the Appendix A. ICU nurses reported the highest workload, which was significantly different from the workload reported by PCU nurses and PCU physicians.

### 3.2. Comparison of Perception of Quality of Interprofessional Collaboration between Palliative Care and Intensive Care Units

All scales showed at least acceptable reliability both among ICU and PCU staff (Cronbach’s α ≥ 0.73, Appendix A). Figure 2 compares the perceived quality of collaboration with different professions between palliative and intensive care physicians and nurses. When considering the overall quality of collaboration with all professions considered, ICU nurses give the lowest ratings of all four groups (*p* ≤ 0.001, Figure 2H). This finding is repeated when looking at the individual professions (Figure 2A–G). The only collaboration for which PCU nurses also gave significantly lower ratings than physicians was collaboration with residents (*p* ≤ 0.05). In general, both ICU and PCU physicians perceived a better quality of collaboration with different professions thannurses and had no significant differences between them.

Figure 3 shows the comparisons regarding inclusive leadership and collaboration in care decisions. Again, intensive care nurses gave the lowest ratings for inclusive leadership by attendings (significant difference from ICU physicians and PCU nurses) and for inclusive leadership by head nurses compared to PCU nurses (*p* ≤ 0.001). For collaboration about care decisions all four groups differed significantly with the perceived quality of collaboration increasing from ICU nurses to ICU physicians, to PCU nurses, and to PCU physicians (*p* ≤ 0.001).

### 3.3. Outcomes Possibly Affected by Quality of Collaboration

The groups showed large differences in the perception of nonbeneficial care with PCU physicians perceiving the lowest frequency, followed by PCU nurses, followed by ICU physicians, and ICU nurses with the highest reported frequency (all groups significantly different from each other, Figure 4A). While there was no significant group difference in emotional exhaustion (Figure 4B), ICU nurses reported the lowest job satisfaction compared to all other groups (Figure 4C). Interestingly, both ICU nurses and PCU physicians reported significantly higher intentions to leave the job compared to PCU nurses (Figure 4D). Further post hoc analysis of intention to leave the job revealed that among PCU physicians only residents but not attendings showed an increased score (Appendix A).

### 3.4. Relation of Aspects of Interprofessional Collaboration with Perception of Nonbeneficial Treatment and Staff Well-Being

Table 2 shows partial correlations, which were adjusted for differences in gender, age, occupation, and workload. There were significant associations of aspects of collaboration with perceived nonbeneficial treatment and staff well-being both for ICU and PCU staff, with some interesting differences. First, the quality of collaboration with nurses was significantly associated with all outcomes among ICU staff, but not among PCU staff. Similarly, the inclusive leadership by head nurses was significantly associated with job satisfaction among ICU nurses, but not among PCU nurses. On the other hand, collaboration with residents and attendings, as well as inclusive leadership by attendings and nurse-physician collaboration about care decisions showed more significant relations to outcomes among PCU staff than among ICU staff. As expected, the perceived frequency of nonbeneficial care was negatively associated with inclusive leadership by attendings and better collaboration about care decisions between physicians and nurses among both ICU and PCU staff. Interestingly, the quality of collaboration with occupational and physical therapists was associated with less perceived nonbeneficial care among ICU staff, while this was the case for the collaboration with psychologists/social workers among PCU staff.

## 4. Discussion

### 4.1. Interpretation of Results

This is the first multicenter study to compare perceptions of the quality of interprofessional collaboration between nurses and physicians in ICUs and PCUs. We increased the validity of the comparison by matching for demographic variables. Overall, PCU staff tended to rate the quality of collaboration with different professional groups higher compared to ICU staff. Compared with ICU physicians, ICU nurses gave lower ratings for the collaboration for five of seven professional groups, while such a difference between nurses and physicians occurred only for PCU staff rating of relationships with residents. In the more detailed assessment of decision-making and leadership, ICU nurses again gave worse ratings than ICU physicians, while there was no respective difference among PCU staff. Thus, our expectations of generally better-working relations in palliative care teams than in critical care teams were confirmed by our results. In addition, ICU nurses reported the most frequent perception of nonbeneficial treatment, the lowest job satisfaction, and the highest intention to leave the job. Controlling demographics and workload in partial correlations, we found that higher quality of collaborative relationships was associated with decreased perception of nonbeneficial care and increased staff well-being. Thus, the collaborative culture of palliative care teams might at least in part explain, why PCU nurses feel much better about their work compared to ICU nurses.

In palliative care, most studies focusing on teamwork have been conducted in the hospice setting [13,15,44], and little is known about collaboration within acute inpatient PCUs. We were able to replicate the finding that palliative care teams have comparably good teamwork in the hospital setting. Since the beginning of the hospice and palliative care movement, there has been a strong emphasis on collaboration in the interprofessional/interdisciplinary care teams and on self-care [13,14]. In contrast to the traditional hierarchies between physicians and other healthcare professionals in hospitals, palliative care teams tend to be non-hierarchical [45]. Quality standards require regular interprofessional team conferences in PCUs, in some settings daily, and in others weekly [46]. These meetings are attended by representatives of different professions (physicians, psychologists, nurses, physical therapists, and spiritual caregivers) to discuss achievable treatment goals and the contribution of each profession to achieving these goals. In ICU care, such team conferences are less established and nurses often do not even participate in patient rounds [47]. Therefore, differences in the perceived quality of interprofessional collaboration could be explained by differences in culture, which are also reflected in aspects of typical structures and processes for coordinating the work of the team.

To our knowledge, this is the first study to measure the frequency of perceived nonbeneficial treatment among palliative care staff. Not surprisingly, PCU staff perceived nonbeneficial treatment less frequently than ICU staff. Perceptions of high-quality collaboration within the team were associated with less perception of nonbeneficial treatment both among ICU and PCU staff. The higher perception of nonbeneficial treatment by ICU nurses compared to physicians has been reported before [25,27,48]. In our survey, PCU nurses also perceived nonbeneficial treatment significantly more often than PCU physicians, which is a new and surprising finding. Although a transfer to the PCU implies a change in the primary goal of treatment from curative to palliative, differences in opinion about the appropriate level of life-sustaining treatment seem to persist within the PCU care team. Within non-curative treatment, the goal of care can differ between prolonging the remaining life span, symptom control, or end-of-life care. Fundamental differences in education, attitudes, and values between nurses and physicians may explain differences in judgments of appropriateness of care in both the ICU and PCU [10], but specific reasons for perceived nonbeneficial treatment were not measured in our study. These may be very different between ICU and PCU staff and deserve future investigation.

There is no conclusive evidence of differences in the risk of burnout between different medical specialties [18,19,49]. In our study, we found no difference between ICU and PCU staff in emotional exhaustion, the central aspect of burnout [50]. Only one previous study compared ICU and PCU staff for burnout and reported twofold odds among critical care staff [22]. Interestingly, this study did not find a difference in the aspect of emotional exhaustion but did find higher depersonalization—a tendency to emotionally distance oneself from patients to cope with stress—and reduced professional accomplishment—the feeling of achieving something worthwhile and meaningful through one’s work—among critical care workers. Therefore, the primary difference might not be in higher levels of distress among ICU staff, but in fewer opportunities to cope positively with distress. Individual training and support to strengthen coping mechanisms are much more common in palliative care and should be adopted in other medical fields [45]. Further research comparing different healthcare settings regarding stressors, supportive conditions, coping strategies, and long-term outcomes such as burnout is needed.

A major issue in health care is the shortage of nurses, particularly in critical care. We found that aspects of a collaborative team environment were associated with increased job satisfaction and decreased intention to leave the job among both ICU and PCU staff. ICU nurses reported the highest intention to leave the job, highlighting the need to improve the quality of their work environment. The relatively high intention to leave the job among palliative care physicians is reported only by junior but not by senior physicians. A likely explanation is the relatively short 6-month rotational training in the PCU that residents must complete to become board-certified in Germany. Therefore, many of the residents will naturally move on to other areas of work after this period.

Although differences in the quality of collaboration between ICU and PCU cannot fully explain the differences in staff well-being, our results suggest that it plays some role. This is consistent with previous research that has repeatedly shown the positive effect of good working relationships on reduced burnout, increased job satisfaction, and decreased intention to quit [4,22,25,27]. Collaborative decision-making and inclusive leadership also prevent the perception of nonbeneficial care, which in turn is associated with moral distress and burnout [25,27,51]. Therefore, improving collaboration within the team could increase well-being and staff retention, especially among ICU nurses. Based on our findings, we propose that the collaborative culture of palliative care can be a role model for improving teamwork in critical care. Interestingly, one intervention study was able to improve collaboration and reduce burnout and depression within the critical care team in the context of end-of-life care [52]—the aspect of intensive care most closely related to palliative care. Changing established cultures and structures is complex and projects to achieve change in healthcare organizations often fail [53]. Leadership behaviors are key to fostering collaboration, effectiveness, learning, and well-being in healthcare teams [32,54]. Inclusive leadership has been shown to overcome traditional status hierarchies among healthcare professionals [32]. We found that the leadership by both attendings and head nurses was perceived as less inclusive by ICU staff compared to PCU staff. Therefore, training to increase the sensitivity and skills of clinician leaders to foster an inclusive and collaborative work climate might be a first step toward improvement.

### 4.2. Strengths, Limitations and Directions for Future Research

The strengths of our study are its multicenter design involving both university and primary care hospitals, the use of validated scales, and the use of a matching approach to control for differences in demographic characteristics in the comparisons between ICUs and PCUs. The study also has limitations. Only German hospitals were included, which limits the generalizability of the results. Due to small within-hospital samples, especially for PCUs, matching could only include a limited number of categorized demographic variables. Using the quantitative method of a survey allowed us to include the perceptions of a larger number of staff from multiple institutions, resulting in better external validity in inferring differences in teamwork between palliative and critical care staff. On the other hand, the standardized nature of quantitative research does not lend itself to a deeper understanding of social processes and meanings. While there are several qualitative studies of teamwork both in palliative and critical care [55,56,57], comparative qualitative studies involving both settings are still lacking. The data are already 8 years old, but no major changes in the structures of palliative and critical care have occurred in Germany since then. The recent COVID-19 pandemic has left its mark on the critical care workforce, as evidenced by increased rates of burnout, anxiety, depression, and post-traumatic stress—especially among ICU nurses [58]. This again highlights the need for changes in the critical care work environment to maintain a healthy and engaged workforce.

## 5. Conclusions

This multicenter survey study showed that many aspects of teamwork are perceived to be worse among ICU staff than among PCU staff, with ICU nurses giving the least favorable ratings. It also showed that less favorable perceptions of collaboration were associated with an increased likelihood of perceiving nonbeneficial care for patients and reporting decreased staff well-being. Critical care could learn from palliative care and strive for a culture of nonhierarchical interprofessional teams, where all healthcare professions are valued for their unique skills, and shared decision-making is facilitated by inclusive clinical leaders. Future studies should extend our findings by seeking a deeper understanding of why there are large differences in the perceived quality of collaboration between palliative and critical care teams. Experiences and shared meanings within teams and involved professional groups could be better understood through qualitative methods such as interviews or ethnographic observations. In addition, innovative quantitative methods such as social network analysis can help to systematically analyze the relationships between different actors in the health care team.

## Figures and Tables

**Figure 1 healthcare-12-00602-f001:**
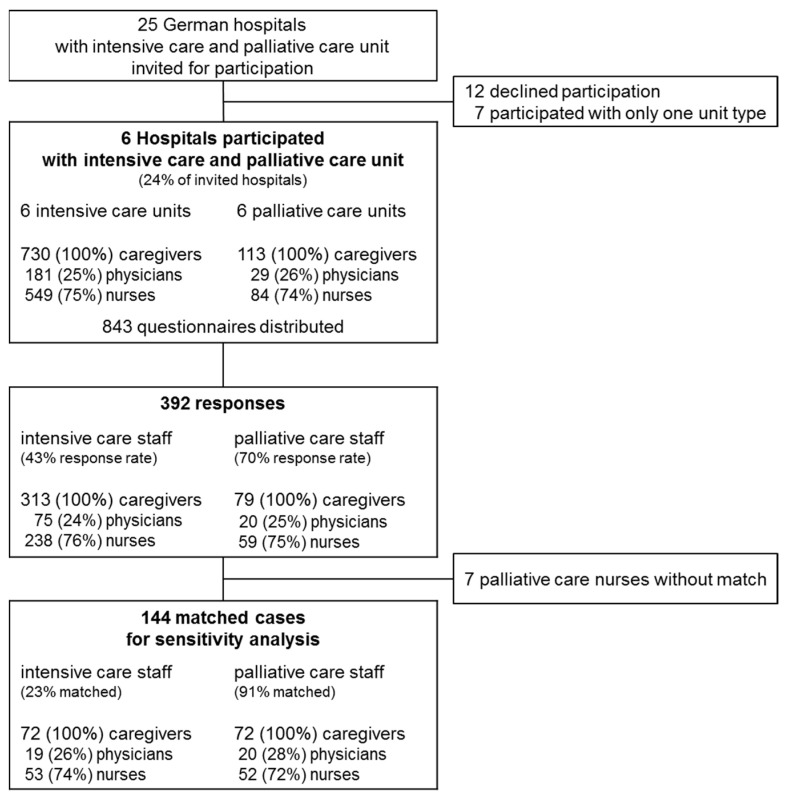
Study flow chart. Matching aims at retaining all cases of the smaller group (palliative care staff), five palliative care nurses were discarded because no appropriate match was found.

**Figure 2 healthcare-12-00602-f002:**
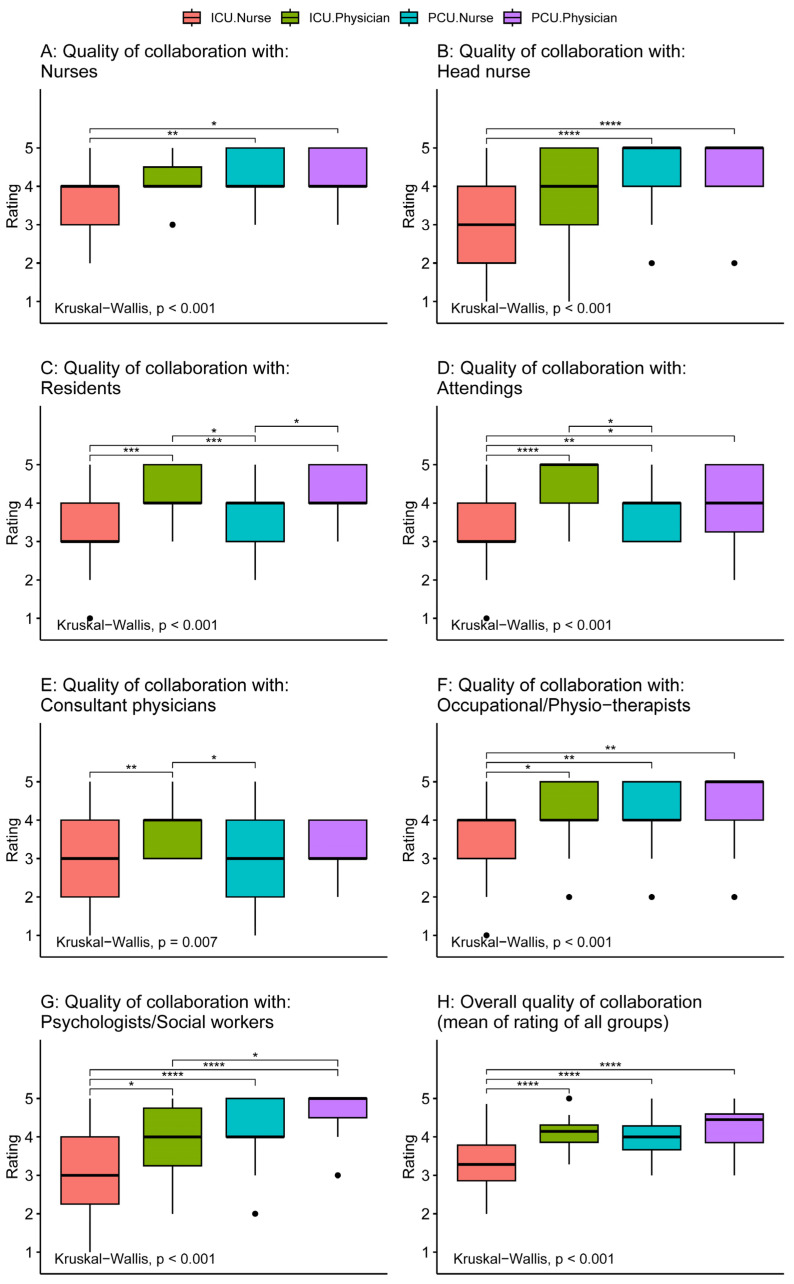
Comparison of perception of quality of interdisciplinary collaboration with different groups between intensive care and palliative care physicians and nurses. Tests for differences between groups were performed by the Kruskal–Wallis test. Comparisons between individual groups were performed using Wilcoxon rank sum tests adjusted for multiple comparisons by the Holm method (significance level: **** *p* ≤ 0.0001, *** *p* ≤ 0.001, ** *p* ≤ 0.01, * *p* ≤ 0.05, no parenthesis: not significant).

**Figure 3 healthcare-12-00602-f003:**
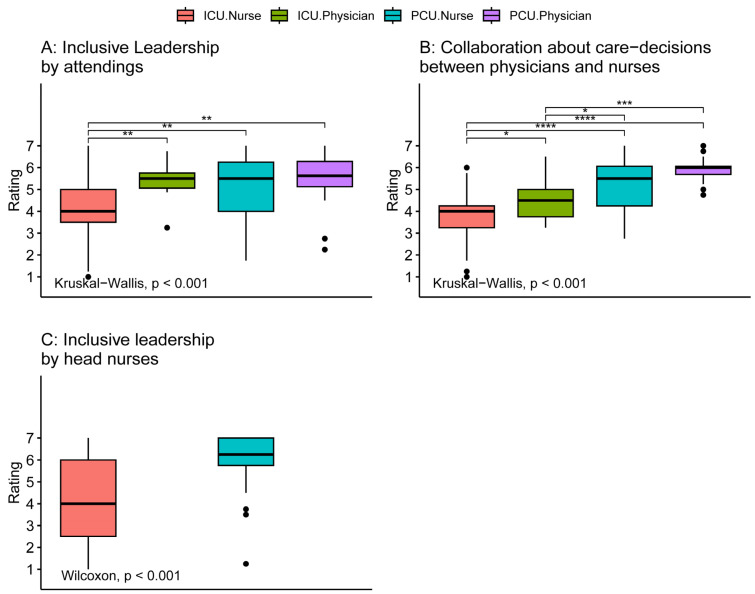
Comparison of perception of inclusive leadership and collaboration about care decisions between intensive care and palliative care physicians and nurses. Tests for differences between groups were performed using the Kruskal–Wallis test. Comparisons between individual groups were performed using Wilcoxon rank sum tests adjusted for multiple comparisons by the Holm method (significance level: **** *p* ≤ 0.0001, *** *p* ≤ 0.001, ** *p* ≤ 0.01, * *p* ≤ 0.05, no parenthesis: not significant). ICU: intensive care unit, PCU: palliative care unit.

**Figure 4 healthcare-12-00602-f004:**
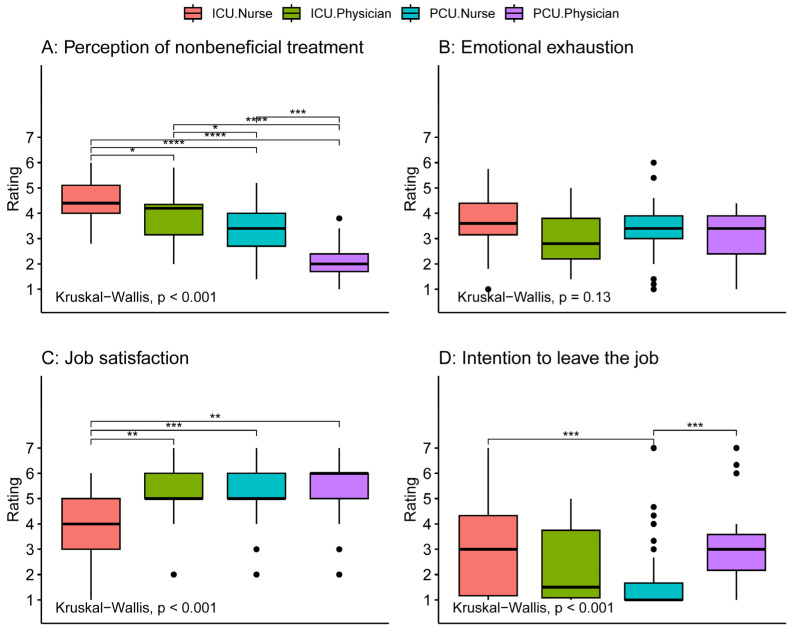
Comparison of perception of non-beneficial care, emotional exhaustion, job satisfaction, and intention to quit between intensive care and palliative care physicians and nurses. Tests for differences between groups were conducted using the Kruskal–Wallis test. Comparisons between individual groups were performed with the Wilcoxon rank sum tests adjusted for multiple comparisons by the Holm method (significance level: **** *p* ≤ 0.0001, *** *p* ≤ 0.001, ** *p* ≤ 0.01, * *p* ≤ 0.05, no parenthesis: not significant). Post hoc analysis of intention to leave the job revealed that among PCU physicians only residents showed an increased rating (Appendix A). ICU: intensive care unit, PCU: palliative care unit.

**Table 1 healthcare-12-00602-t001:** Demographic characteristics of participating staff.

Variable	Before Matching			After Matching ^1^		
	ICU, N = 313	PCU, N = 79	*p* Value	ICU, N = 72	PCU, N = 72	*p* Value
Gender: Female ^2^	209 (69.9)	55 (74.3)	0.48	49 (71)	48 (71.6)	1
Age (years) ^2^: <30	112 (37.7)	5 (6.9)	≤0.001	9 (13)	5 (7.7)	0.262
30–39	108 (36.4)	14 (19.4)		15 (21.7)	14 (21.5)	
40–49	59 (19.9)	31 (43.1)		34 (49.3)	27 (41.5)	
≥50	18 (6.1)	22 (30.6)		11 (15.9)	19 (29.2)	
Medical experience (years) ^2^: <1	8 (2.8)	1 (1.4)	≤0.001	0 (0)	1 (1.6)	0.511
1–2	29 (10.2)	3 (4.2)		1 (1.6)	3 (4.7)	
3–5	57 (20.1)	2 (2.8)		5 (7.8)	2 (3.1)	
6–10	55 (19.4)	8 (11.3)		8 (12.5)	8 (12.5)	
>10	134 (47.3)	57 (80.3)		50 (78.1)	50 (78.1)	
Experience in intensive/palliative care (years): <1	29 (10.3)	6 (8.6)	≤0.001	3 (4.8)	6 (9.5)	≤0.001
1–2	41 (14.6)	9 (12.9)		3 (4.8)	9 (14.3)	
3–5	63 (22.4)	26 (37.1)		9 (14.3)	23 (36.5)	
6–10	48 (17.1)	24 (34.3)		12 (19)	21 (33.3)	
>10	100 (35.6)	5 (7.1)		36 (57.1)	4 (6.3)	
Job role ^2^: Nurse	232 (74.1)	53 (67.1)	0.01	50 (69.4)	49 (68.1)	0.995
Head nurse	6 (1.9)	6 (7.6)		3 (4.2)	3 (4.2)	
Junior physician	52 (16.6)	9 (11.4)		8 (11.1)	9 (12.5)	
Senior physician	23 (7.3)	11 (13.9)		11 (15.3)	11 (15.3)	

Descriptive data are presented as n (%) or median [1st quartile, 3rd quartile]. Significance testing was executed by Fisher’s exact test, Chi-squared test, or Mann-Whitney U test, as appropriate. ICU: intensive care unit. PCU: palliative care unit. ^1^ 1 to 1 matching stratified by hospital using a genetic matching algorithm to optimize balance between unit types within each hospital [41]. ^2^ Characteristics considered for matching with age dichotomized (<40 or ≥40 years) and medical experience dichotomized (<5 or ≥5 years; see Appendix A for standardized differences before and after matching).

**Table 2 healthcare-12-00602-t002:** Relation between quality of collaboration, perception of nonbeneficial care, and staff well-being.

	Ratings by ICU Staff				Ratings by PCU Staff			
	Perception of Nonbeneficial Care	Emotional Exhaustion	Job Satisfaction	Intention to Quit	Perception of Nonbeneficial Care	Emotional Exhaustion	Job Satisfaction	Intention to Quit
Quality of collaboration with: Nurses	−0.31 *	−0.3 *	0.46 ***	−0.27 *	0.1	0.06	0.06	−0.13
Quality of collaboration with: Head nurse	−0.24	−0.19	0.5 ***	−0.16	−0.09	0.09	0.13	−0.05
Quality of collaboration with: Residents	−0.11	−0.11	0.4 **	−0.1	−0.34 *	0.01	0.26	−0.34 *
Quality of collaboration with: Attendings	−0.21	−0.05	0.44 ***	−0.02	−0.36 **	−0.34 *	0.52 ***	−0.32 *
Quality of collaboration with: Consultant physicians	0.13	0.04	0.35 **	0.1	−0.09	−0.12	0.31 *	−0.23
Quality of collaboration with: Occupational/Physio-therapists	−0.35 **	−0.14	0.07	−0.09	0	−0.11	0.29 *	−0.04
Quality of collaboration with: Psychologists/Social workers	−0.11	0.07	0.1	−0.03	−0.37 **	0.11	0.2	−0.01
Inclusive leadership by attendings	−0.31 *	−0.2	0.31 *	0.03	−0.28 *	−0.48 ***	0.57 ***	−0.3 *
Collaboration about care-decisions between physicians and nurses	−0.4 **	−0.14	0.37 **	−0.07	−0.45 ***	−0.31 *	0.32 *	−0.27 *
Inclusive leadership by head nurse ^1^	−0.29	−0.13	0.42 **	−0.18	0	−0.12	−0.05	−0.02

Table shows partial Spearman’s correlations controlling for occupation (nurse vs. physician), gender, age (<40, ≥40), and workload. ICUs: intensive care units. PCUs: palliative care units. *** *p* ≤ 0.001, ** *p* ≤ 0.01, * *p* ≤ 0.05. ^1^ Inclusive leadership by the head nurse was assessed and analyzed only for nurses of PCUs and ICUs; therefore, the partial correlations were only controlled for gender, age, and workload.

## Data Availability

The data presented in this study are available on reasonable request from the corresponding author in anonymized form after approval by the data protection officer of the Jena University Hospital.

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
