# Peer review of "Perceptions of Quality of Interprofessional Collaboration, Staff Well-Being and Nonbeneficial Treatment: A Comparison between Nurses and Physicians in Intensive and Palliative Care"

_healthcare, 2024, doi:10.3390/healthcare12060602_

Round 1
Reviewer 1 Report
Comments and Suggestions for Authors
Dear authors, I find the text you propose interesting for what is being investigated. I propose a series of considerations with the aim of improving its quality:
-When the purpose of a study is to find out the perception on a specific topic, the most appropriate instruments are those related to qualitative methodology, rather than questionnaires. This could constitute a considerable bias, as participants do not freely express their opinions, but answer closed questions. It is recommended to explain the reasons why a qualitative approach is not used in this research and to suggest possible solutions to this bias. -The authors develop an ad hoc scale, but do not assess its psychometric quality, nor do they explain the indications of the quality of the scales used, or where they have been used before.-The study design and sample collection are not sufficiently explained.
Author Response
Dear reviewer,
Thank you very much for your thoughtful comments. Based on these, we revised the manuscript. Please find our point-by-point reply in the following.
Dear authors, I find the text you propose interesting for what is being investigated. I propose a series of considerations with the aim of improving its quality:
-When the purpose of a study is to find out the perception on a specific topic, the most appropriate instruments are those related to qualitative methodology, rather than questionnaires. This could constitute a considerable bias, as participants do not freely express their opinions, but answer closed questions. It is recommended to explain the reasons why a qualitative approach is not used in this research and to suggest possible solutions to this bias. –
Answer: Dear reviewer, thank you very much for your comment. We agree that qualitative research due to its openness to subjective experience has certain advantages compared to quantitative research by standardized questionnaires. At the same time, we do not agree that qualitative research is the only way to assess perceptions on collaboration and teamwork. This is indicated by numerous studies, which use survey instruments to assess teamwork and collaboration in health care as well as numerous surveys in management science or organizational psychology. Both approaches have their own unique advantages and neither one can generally be preferred over the other. We added considerations regarding quantitative and qualitative research to the discussion of limitations:
“Using the quantitative method of a survey allowed us to include the perceptions of a larger number of staff from multiple institutions, which results in better external validity in inferring differences in teamwork between palliative and critical care staff. On the other hand, due to its standardized nature, quantitative research is not suited to allow a deeper understanding of social processes and meaning. While there are several qualitative studies on collaboration within the team both in palliative and critical care [52-54], comparative qualitative studies involving both fields are still missing.”
The authors develop an ad hoc scale, but do not assess its psychometric quality, nor do they explain the indications of the quality of the scales used, or where they have been used before.
Answer: We do not agree that our questionnaire should be regarded an ad hoc instrument, since it is based on a structured development process and relies on other pre-existing and validated instruments. We already gave a citation for each of the previously published scales. We now extended the description of the development of the questionnaire at the beginning of the “Measurement” paragraph in the methods section:
“2.2.1. Development of the questionnaire
The questionnaire was developed in a process involving several steps. First, a theoretical framework was developed including three broad themes a) perception of the quality of collaboration on the unit, b) relevant outcomes affected by the quality of collaboration (staff well-being, perception of nonbeneficial care), c) relevant covariates (demographic characteristics, workload). Second, since no single validated questionnaire assessing all relevant constructs existed, we conducted a literature search to select relevant, validated scales or items from existing instruments. Item wordings were partly adapted to the setting of the study (e.g. referring to nurses/physicians or the unit instead of co-workers or the company), English items were translated to German by forward and backward transla-tion. No validated scale existed to measure perception of nonbeneficial treatment. Therefore, we developed a new scale. This initial pool of items was pretested by cognitive interviewing with nurses and physicians from critical and palliative care. The reliability and factorial validity of the scales was tested in the larger sample of 23 ICUs and was reported previously [26]. To assure reliability of the scales also among PCU staff, we reassessed it in the sample of PCU nurses and physicians.”
-The study design and sample collection are not sufficiently explained.
Answer: Thanks for your advice. We have added a description of the study design as the first paragraph of the methods section:
“This is a prospective observational study conducted by a paper pencil survey of nurses and physicians of both ICUs and PCUs in a convenience sample of six German hospitals. To adjust for differences in demographic characteristics between ICU and PCU staff, a 1-to-1 matching was conducted within each participating hospital.”
In addition, we extended the description of the sample collection and procedure of the survey:
“This is a sub study conducted in the context of a cluster randomized controlled trial, which aimed to improve acute sepsis care (Medical Education for Sepsis Source Control and Antibiotics, MEDUSA, ClinicalTrials.gov Identifier NCT01187134) [27]. All participating hospitals were invited to take part in a survey of ICU staff, of which results have been reported previously [26]. Twenty-five of the hospitals in the MEDUSA trial had both a PCU and an ICU and were invited to participate in a paper pencil survey of both palliative care and critical care nurses and physicians; six of these hospitals took part in the survey. Experienced senior physicians of heads of department served as local study coordinators and distributed the paper pencil survey to the staff of the units between December 2013 and March 2015. All physicians and nurses working on the respective PCUs and ICUs were eligible for participation. Each local study coordinator received feedback on the rate of participation within his unit after two and four weeks, and questionnaires were distributed among staff a second time after four weeks.”
Reviewer 2 Report
Comments and Suggestions for Authors
The authors address a topic of potential high practical impact.
The authors provide a very concise summary of the results from the literature to orient readers to their questions.
Line 34. Details on findings from studies referred to here (3-7) can effectively support readers in developing a more detailed representation of the relevance of the area investigated.
Line 40 and following: same as before referring to studies 13,15, 16.
Line52. Please insert the full definition of the acronym PCU here.
Lines 54-56. Here authors describe the aims of the study. The theoretical background or perspectives behind their choices are not mentioned here. I refer, for instance to the model or theoretical reference underlying wellbeing that guides the choice of the specific tools they used in the study. Making these choices explicit can provide authors also reflections and comments to add to the discussion and the conclusion section.
From line 66 authors describe the measures used. This section should be more clearly organized to help readers identify the diverse areas addressed in the study. It may be useful for instance to use italics to identify the three sections and the subcomponents addressed. This will also parallel the description of the results that the authors provide.
Page 7. Figure 3 proposes a summary of some results. The authors put them together here because they refer to a closer area of analysis. Finding a heading grouping them here and in the first description of the measures could help authors underline the usefulness and meaning of the protocol they developed for further studies. It may also be worth describing the results in a distinct paragraph. They refer more to work tasks and functions (including decision-making).
The same line of thinking could be interesting ( but more secondary) for the dimensions in paragraph 3.3.
As regards discussion, at the very beginning authors should more clearly describe the main findings and organize them so that readers can parallel the findings with the results and according to the sections of the protocols, i.e. the dimension addressed.
Line 253. Authors use the word “conferences” which could be seen as a too broad term, given the specific context it refers to. If they still use it, please specify the meaning and the way it is used in this context.
Lines 265 and following. The analysis of the results referring to wellbeing could be more extensively described with a model of wellbeing in mind (see suggestion provided earlier) thus underlining the relevance of the findings and potentially opening to comments on preventive/ early protective actions to pursue ( as it is someway anticipated in the abstract).
Finally, based on suggestions, in the conclusions, authors may focus on the potential use of the protocol they assembled for further studies.
Comments on the Quality of English LanguageA general check of the English is suggested for more attention to detail.
Author Response
Dear reviewer,
Thank you very much for your thoughtful comments. Based on these, we revised the manuscript. Please find our point-by-point reply in the following.
The authors address a topic of potential high practical impact.
The authors provide a very concise summary of the results from the literature to orient readers to their questions.
Answer: Thank you very much.
Line 34. Details on findings from studies referred to here (3-7) can effectively support readers in developing a more detailed representation of the relevance of the area investigated.
Answer: We included a summary of the major findings of the cited studies:
“High quality collaboration is associated with improved patient outcomes and health care staff well-being. For example, ICU nurses perception of good nurse-physician collaboration about transfer decisions from ICU has been found to be associated with decreased risk of patients’ mortality or re-admission to ICU [3]; the rating of nurse-physician relations as well as nurse management were associated with less experience of burnout, higher job satisfaction, and improved assessment of quality of care by hospital nurses [4]; better collaboration in the interprofessional team of a surgical unit was associated with improved quality of care and postoperative functioning of patients as well as reduced pain and length of stay [5]; better rating of nurse-physician collaboration assessed by nurses of surgical and medical wards was associated with reduced length-of-stay of patients [6]; a literature review on team approaches in palliative care concluded that communication, leadership skills and mutual respect are key to the success of multidisciplinary teams [7].”
Line 40 and following: same as before referring to studies 13,15, 16.
Answer: we extended the description of previous studies on collaboration in palliative care teams:
“Palliative care has a long tradition of emphasizing the interprofessional/interdisciplinary team and is regarded a working model for successful collaboration [13,14]. Therefore, it can be expected that palliative care teams give generally better ratings of collaboration, which are less different between different professions. Accordingly, two survey studies among hospice teams in the US revealed no differences between the professions (e.g. nurses, physicians, social workers, and chaplains) in their rating of the quality of interdisciplinary collaboration [13,15]. Beside small single-center studies, little is known about interprofessional/interdisciplinary collaboration in inpatient palliative care units (PCUs) [16,17]. We conducted a single-center study comparing experiences in the context of end-of-life decision making between ICU and PCU staff, where we found that PCU staff gave better ratings of the interaction within the team, and while ICU nurses rated interaction worse than ICU physicians, ratings of PCU nurses and physicians did not differ [16].”
Line52. Please insert the full definition of the acronym PCU here.
Answer: we now introduce the acronym at the first mentioning of palliative care units.
Lines 54-56. Here authors describe the aims of the study. The theoretical background or perspectives behind their choices are not mentioned here. I refer, for instance to the model or theoretical reference underlying wellbeing that guides the choice of the specific tools they used in the study. Making these choices explicit can provide authors also reflections and comments to add to the discussion and the conclusion section.
Answer: We did not strive to test a specific theory of interdisciplinary collaboration, but aimed for investigating specific questions comparing the critical and palliative care setting. We extended the explanation of our study aims to better show, how they connect to the current evidence presented in the introduction:
“Based on the cited literature, we assume that the quality of collaboration is an essen-tial predictor for staff well-being and for the prevention of provision of nonbeneficial treatment both in critical and palliative care. We further assume that palliative care teams show a better quality of collaboration compared to critical care teams with less differences between professions in the ratings of quality of collaboration. To investigate this, we con-ducted a multi-center survey study among staff of ICUs and PCUs.”
From line 66 authors describe the measures used. This section should be more clearly organized to help readers identify the diverse areas addressed in the study. It may be useful for instance to use italics to identify the three sections and the subcomponents addressed. This will also parallel the description of the results that the authors provide.
Answer: We revised the structure of this paragraph and introduce the different scales and items under the headings a) perception of quality of collaboration, b) outcomes possibly affected by quality of collaboration, and c) Covariates (which means workload and demographics).
Page 7. Figure 3 proposes a summary of some results. The authors put them together here because they refer to a closer area of analysis. Finding a heading grouping them here and in the first description of the measures could help authors underline the usefulness and meaning of the protocol they developed for further studies. It may also be worth describing the results in a distinct paragraph. They refer more to work tasks and functions (including decision-making).
Answer: Thank you for this comment. Actually collaborative decision-making and leadership should encompass aspects of effective collaboration in the team (see above). Therefore, workload does not really fit here – we assessed it as a covariate and not a variable of primary interest. To better structure the presentation of our results, we therefore excluded workload from Figure 3. Differences in Workload are now presented in the online supplement and referred to in the first paragraph of the results.
The same line of thinking could be interesting ( but more secondary) for the dimensions in paragraph 3.3.
Answer: we changed the title of this paragraph to better fit it to the revised description of our measures: “3.3. Outcomes possibly affected by quality of collaboration“.
As regards discussion, at the very beginning authors should more clearly describe the main findings and organize them so that readers can parallel the findings with the results and according to the sections of the protocols, i.e. the dimension addressed.
Answer: we restructured the first paragraph of the discussion to give a better summary of our results:
“This is the first multicenter study that compares perceptions of the quality of interprofessional collaboration between nurses and physicians on ICUs and PCUs. We increased the validity of the comparison by matching for demographic variables. Overall, PCU staff tended to rate collaboration with different professional groups better compared to ICU staff. Compared to ICU physicians, ICU nurses gave worse ratings for the collaboration for five of seven professional groups, while such difference between nurses and physicians occurred only for the rating of relations to residents among PCU staff. Considering the more detailed assessment of decision-making and leadership, again ICU nurses gave worse ratings than ICU physicians did, while there was no respective difference among PCU staff. Thus, our expectations regarding generally better collaborative relations in palliative care teams compared to critical care teams were confirmed by our results. In addition, ICU nurses reported the most frequent perception of nonbeneficial treatment, had the lowest job satisfaction and highest intention to leave the job. Controlling for demographics and workload in partial correlations, we found that higher quality of collaborative relations was associated with decreased perception of nonbeneficial care and increased staff well-being. Thus, the collaborative culture of palliative care teams might at least in part explain, why PCU nurses feel much better about their work compared to nurses in critical care.“
Line 253. Authors use the word “conferences” which could be seen as a too broad term, given the specific context it refers to. If they still use it, please specify the meaning and the way it is used in this context.
Answer: We extended the description of the conferences:
“Quality standards request regular interprofessional team conferences in PCUs, in some settings daily, others weekly [45]. These meetings are attended by representatives from different professions (doctors, pyschologists, nurses, physiotherapists, spiritual care givers) and discuss achievable treatment aims and contribution of each profession to reach the aims.”
Lines 265 and following. The analysis of the results referring to wellbeing could be more extensively described with a model of wellbeing in mind (see suggestion provided earlier) thus underlining the relevance of the findings and potentially opening to comments on preventive/ early protective actions to pursue ( as it is someway anticipated in the abstract).
Answer: We have now added a paragraph integrating our findings on collaboration and staff wellbeing and giving comments on actions to improve collaboration and wellbeing in intensive care. For a better structure of the discussion, we changed the position of the paragraph on perception of nonbeneficial care and put it in front of the discussion of staff well-being:
“While differences in the quality of collaboration between ICU and PCU cannot completely explain the differences in staff well-being, our results indicate that it plays a certain role. This is in line with previous research, which has repeatedly shown the positive effect of good working relations on reduced burnout, increased job satisfaction, and decreased intention to quit [4,22,24,26]. Collaborative decision making and inclusive leadership do also prevent perception of nonbeneficial care, which itself is associated to moral distress and burnout [24,26,49]. Therefore, improving collaboration within the team could increase well-being and staff retention especially among ICU nurses. Based on our results, we propose that the collaborative culture of palliative care can be a role model to improve teamwork also in critical care. Interestingly, one intervention study was able to improve collaboration and reduce burnout and depression within the critical care team in the context of end-of-life care [50] – the aspect of intensive care most closely related to palliative care. Changing established cultures and structures is complex and projects to achieve change in health care organizations often fail [51]. Leader’s behavior is key in fostering collaboration, effectiveness, learning, and well-being in health care teams [31,52]. Inclusive leader-ship has been shown as a means to overcome traditional status hierarchies between health care professions [31]. We found that leadership both by attendings and head nurses was perceived as being less inclusive by ICU staff compared to PCU staff. Therefore, trainings to increase sensibility and skills of clinician leaders for fostering an inclusive and collaborating work climate might be a first step to improvement.”
Finally, based on suggestions, in the conclusions, authors may focus on the potential use of the protocol they assembled for further studies.
Answer: From our point of view, further survey research could add only minor new insights to help developing preventive strategies. We therefore propose gaining deeper insights into differences in the quality of collaboration between palliative and critical care using different study designs. We added this to the conclusions section:
“Future studies should extend on our findings by striving for a deeper understanding on why the large differences in the perceived quality of collaboration between palliative and critical care teams occur. Experiences and shared meaning within teams and involved professional groups could be more deeply understood by qualitative methods, like interviews or ethnographic observations. Also innovative quantitative methods like social network analysis can help to systematically analyze relations between different actors in the health care team.”
Comments on the Quality of English Language
A general check of the English is suggested for more attention to detail.
Answer: We will have a professional translator check the revised manuscript.
Reviewer 3 Report
Comments and Suggestions for Authors
One surprising aspect of this contribution is the timing...I understand this has to do with the translation of a German study? On 311 you adequately refer to this, but I propose you should also mention this in the abstract and the introduction.
Author Response
Dear reviewer, the delay of publications of the results is due to other obligations of the first author after the gathering of the data. We agree that this is a limitation of our results. We added the dates of the conduction of the survey to abstract and introduction.
Reviewer 4 Report
Comments and Suggestions for Authors
Dear authors,
Thank you for your work on this interesting topic. As a physician who has worked in both intensive care and palliative care, I can very well understand the results presented. It is a real shame that the structures of palliative care are not also used in intensive care, although anyone who knows both areas will recognise the advantages. Therefore it is a pity that you have not published the data until now. Although you write that nothing has changed in the medical disciplines since the study was conducted, I believe that the COVID pandemic has created an even bigger problem here, especially in intensive care. Perhaps this could still be discussed?
Other than that, I just have a few minor comments that I'm sure will be resolved quickly:
1. Please use the English abbreviations (ICU, PCU) you also use in the text in Table 1 instead of the German abbreviations (ITS, Palli)
2. In table 2 there is only a small beauty problem with the formatting. In the first column, you could always put a paragraph after the :
3. According to the new "Weiterbildungsordnung" from 2018, only 6 instead of 12 months are required for board certification in palliative medicine. Please change this in the discussion.
4. Perhaps you could consider writing a separate section on strengths and limitations instead of including this in the discussion.
5. The DIVI position paper on overtreatment in intensive care medicine should be used as a reference, as we know that non-beneficial care is a real problem in intensive care medicine and is not only perceived as such by nursing staff (doi: 10.1007/s00063-021-00794-4).
Author Response
Dear reviewer,
Thank you very much for your thoughtful comments. Based on these, we revised the manuscript. Please find our point-by-point reply in the following.
Dear authors,
Thank you for your work on this interesting topic. As a physician who has worked in both intensive care and palliative care, I can very well understand the results presented. It is a real shame that the structures of palliative care are not also used in intensive care, although anyone who knows both areas will recognise the advantages. Therefore it is a pity that you have not published the data until now. Although you write that nothing has changed in the medical disciplines since the study was conducted, I believe that the COVID pandemic has created an even bigger problem here, especially in intensive care. Perhaps this could still be discussed?
Answer: Thank you very much for this important advice. We added this topic at the end of the discussion, including a new reference (Wahlster S, Hartog C. Coronavirus disease 2019 aftermath: psychological trauma in ICU healthcare workers. Curr Opin Crit Care. Dec 1 2022;28(6):686-694. doi:10.1097/mcc.0000000000000994):
“The recent Covid-19 pandemic left its marks on critical care professionals represented in increased rates of burnout, anxiety, depression, and post-traumatic stress – especially among ICU nurses [56]. This again highlights the necessity of changes of the work envi-ronment in critical care to retain a healthy and engaged workforce.”
Other than that, I just have a few minor comments that I'm sure will be resolved quickly:
- Please use the English abbreviations (ICU, PCU) you also use in the text in Table 1 instead of the German abbreviations (ITS, Palli)
Answer: Thank you, we corrected this.
- In table 2 there is only a small beauty problem with the formatting. In the first column, you could always put a paragraph after the :
Answer: Done.
- According to the new "Weiterbildungsordnung" from 2018, only 6 instead of 12 months are required for board certification in palliative medicine. Please change this in the discussion.
Answer: Thank you very much for pointing this out. We corrected this.
- Perhaps you could consider writing a separate section on strengths and limitations instead of including this in the discussion.
Answer: we included respective subheadings to the Discussion section.
- The DIVI position paper on overtreatment in intensive care medicine should be used as a reference, as we know that non-beneficial care is a real problem in intensive care medicine and is not only perceived as such by nursing staff (doi: 10.1007/s00063-021-00794-4).
Answer: we included the reference to the introduction:
“Provision of nonbeneficial care at the end-of-life causes unnecessary risk and suffering for patients and families as well as a waste of health care resources [24].”
Round 2
Reviewer 1 Report
Comments and Suggestions for Authors
Dear authors, following the modifications made to the text and the response to comments, the quality of the text has improved. You have responded to the questions and doubts raised by this reviewer.
Author Response
Thank you very much.